# Socio-Economic Predictors of Hiring Live-In Migrant Care Workers to Support Community Dwelling Older Adults with Long-Term Care Needs: Recent Evidence from a Central Italian Region

**Oliver Fisher** [1,2,*] **, Paolo Fabbietti** [3] **and Giovanni Lamura** [1]

1 Centre for Socio-Economic Research on Ageing, IRCCS INRCA-National Institute of Health and Science on Ageing, 60124 Ancona, Italy; g.lamura@inrca.it
2 Department of Economics and Social Sciences, Università Politecnica delle Marche, 60121 Ancona, Italy
3 Unit of Geriatric Pharmacoepidemiology, IRCCS INRCA-National Institute of Health and Science on Ageing, 60124 Ancona, Italy; p.fabbietti@inrca.it
* Correspondence: o.fisher@inrca.it

**Abstract:** To meet the rising demand for home care, many families in Italy hire live-in migrant care workers (MCWs). However, the reliance on MCWs to provide long-term care (LTC) and a lack of alternative formal care services raises concerns around equality in access to care. This study aimed to determine the socio-economic predictors of hiring live-in MCWs among older adults with LTC needs in Italy, the objective care burden placed on MCWs, and the financial barriers that people in need of care and informal caregivers face when hiring MCWs, analysing data from a cross-sectional questionnaire with 366 older adults with LTC needs and their primary family caregivers living in the Marche region. Binary logistic regression was used to calculate the predictors of hiring a live-in MCW. Having a primary caregiver that had a high school education or above significantly increased the odds of hiring a live-in MCW (Odds Ratio (OR) = 3.880), as did receiving a social pension (OR = 2.258). Over half (57.5 percent) of the people in need of care had difficulties in affording the costs of hiring an MCW in the past year. To increase the sustainability of the Italian MCW market and reduce socio-economic barriers to accessing care, the Italian Government should increase funding for LTC benefits and add means testing and restrictions on the use of cash-for-care allowances.

**Keywords:** home-care; Italy; migrant care work; long-term care

## 1. Introduction

### 1.1. Long-Term and Informal Care in Italy

The demand for home care in Italy is rising, in part due to its ageing demographic. Italy has the highest proportion of older adults in Europe. In 2020, it was estimated that 23.3 percent of the population was over the age of 65, while the number of people aged 85 years or older increased by 80 percent between 2005 and 2020 [1]. Furthermore, the number of informal caregivers is estimated to decrease in coming years due to increases in women's labour force participation, rising old-age dependency ratios, and drops in intergenerational co-residency [2–4].

The Italian care regime is represented by a strong degree of familism. Traditionally, family members, often women, were responsible for filling the demand for long-term care (LTC) and home care in Italy [5–8]. It was estimated that 16.4 percent of the Italian population aged 15 years or older provide informal care, with 85.4 percent of these informal caregivers providing care for a family member [9,10]. Another study by Tur-Sinai et al. [11] estimated that between 13.66 and 20.02 percent of adults 50 years or older are informal caregivers.

While the care that informal caregivers provide is crucial, it can also harm their wellbeing. Past research showed that the stress of providing care alongside competing demands

for paid work can result in informal caregivers having lower wellbeing, increased cardio-vascular problems, and higher rates of depression than those not providing care [12–15].

The high burden on informal caregivers is further exasperated in Italy due to the low level of formal LTC services available to people in need of care, which were further reduced in recent years due to fiscal austerity measures by the Italian state, both at national and regional levels [7]. In 2017, it was estimated that only 2.1 percent of older adults used residential care services [16], while in the Marche region, where this study took place, only 1.9 percent of older adults used residential care services [16].

*1.2. The Italian Migrant Care Worker Market*

To meet the rising demand for home care, many families have turned to hire live-in migrant care workers (MCWs) [17]. Consequently, there has been a shift away from the traditional family model of care to a migrant-in-the-family model [17]. From the Italian government's perspective, the rise of this market was viewed as a low-cost solution to providing LTC, with the majority of costs being born by people in need of care and their family members [18–20]. From the individual and family level, the creation of this market helped to maintain the preference of older adults to receive care within the home setting [21–23].

In 2019, there were 848,987 care and domestic workers in Italy with formally registered contracts, of which 596,964 (70.3 percent) were migrants [24]. More than 50 percent of these workers were from Eastern Europe, with the next most common regions of origin being South America and Northern Africa. Around half of these workers (48 percent) were employed on care worker contracts, with a further 52 percent employed as domestic workers [24].

MCWs with formal contracts that provide care for older adults with LTC needs are required to be paid between EUR 997.61 and EUR 1232.33 per month, depending on their level of skills training and tasks [25]. Workers are expected to be employed for 54 h a week and no more than 10 h per day [26]. Overtime hours are allowed, but employers are required to pay overtime rates [26]. However, in practice, these maximum hours are rarely adhered to. In part due to the undervaluation of care work, employers often expect MCWs to be working around the clock. This often leaves workers with little or no time off working excessive hours over the hours stipulated in the work contracts [27–29]. Moreover, MCWs employed outside of the formal sector are unlikely to hold any form of employment contract [30].

The Italian MCW market is characterised by a high degree of informality. Complex immigration procedures often result in MCWs from outside the European Union entering Italy through irregular channels, including, for example, by entering on tourist visas or visa exemptions [19]. These MCWs have often relied on ad hoc regularisation processes to gain a regular migration status [8,21,27]. The most recent of these regularisation campaigns occurred between 1 June 2020 and 15 August 2020, resulting in 122,247 applications [31]. It was estimated that in 2020 only 40 percent of MCWs were hired through regular channels [10].

The emergence of the MCW market in Italy can partially be attributed to the use of unregulated cash-for-care allowances [32]. The indennità di accompagnamento (IdA) is the only allowance granted at the national level and is a universal, non-means-tested allowance granted to people with a disability that makes them unable to carry out daily activities without continuing assistance from others [33]. As of 2021, those eligible receive EUR 522.10 per month, which increases to EUR 938.35 if the person is blind [33,34]. In 2018, 11.8 percent of older adults received the IdA [10].

The IdA does not have any conditions attached to its use, with recipients able to decide how the money is spent. Hiring an MCW through formal channels is not feasible for some, as the IdA only covers some of the cost of hiring an MCW [6]. Consequently, as a way to cut care costs and to avoid paying social security contributions, many people have used the IdA to hire MCWs outside of the formal economy [8,21]. This can lead to several issues for MCWs themselves since those with an irregular migration status have

fewer avenues for reporting labour rights violations, do not receive a pension, have limited access to health care, and may fear arrest or deportation [23,27,35,36].

Apart from the IdA, there are some allowances and in-kind services that are granted at the regional level and administered at the municipal level. Additionally, in 2012, the National Institute for Social Security created the Home Care Premium scheme, which includes a cash-for-care allowance that covers the cost of employing a live-in care worker with a regular employment contract, and second, the provision of in-kind services by local municipalities. This scheme was found to increase the formality of the MCW market; however, it is currently only available to public sector employees or their family members with low income [18].

### 1.3. Behavioural Model of Health Service Use

Given the concerns around the affordability of hiring a live-in MCW in Italy, it is important to understand which factors may increase the odds of being able to hire a worker. From a conceptual standpoint, the Andersen health care utilisation model, also known as the behavioural model (BM; see Figure 1), can be used to see which factors may influence access to care services [37]. The BM stresses that improving access to health care is best accomplished when focusing on and understanding contextual and individual determinants of access to care. Contextual factors are measured at the aggregate level and range from units as small as the family level to those as large as national health care systems. Both the contextual and individual levels can be broken up into predisposing, enabling, and needs factors [37]. This article mainly focused on factors at the individual level.

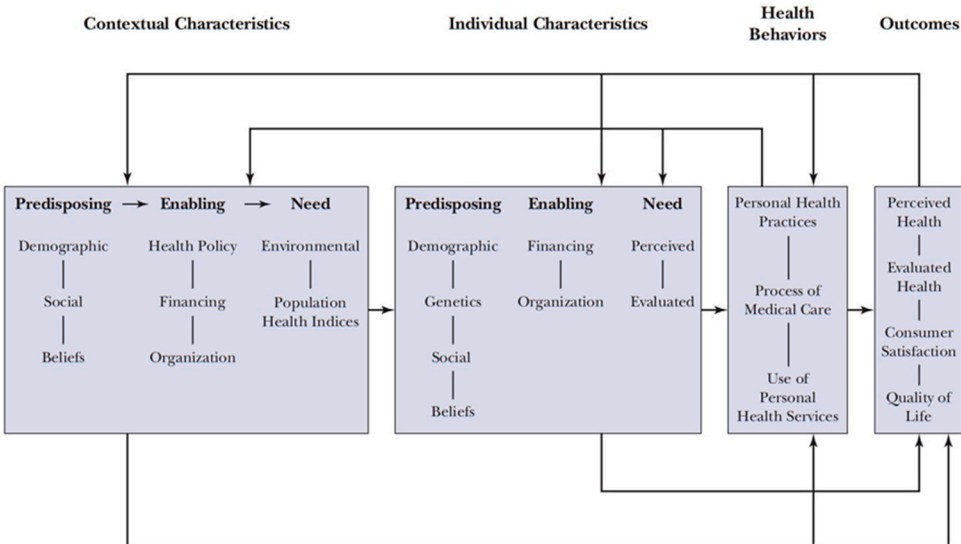

**Figure 1.** Behavioural model of health services use. Reprinted from Andersen and Davidson (2007) [37].

Predisposing characteristics include demographic information such as age, gender, and marital status. Social factors focus on the social status of a person within a community and include education, occupation, and ethnicity. Health beliefs detail how attitudes and values about health services influence the perception of the need and use of health care services [37].

Enabling characteristics focus on the financing of health services and the income available to pay for services. This can include income levels and the costs of care services. The organisation of health care includes the means of travel needed to receive care and the reported travel and waiting time [37].

Needs characteristics include how individuals perceive their health and functional status. This also covers how people experience and emotionally respond to pain or illness. Likewise, this includes how individuals determine when and if to seek care and how this is

explained by social characteristics and health beliefs. Evaluated need represents objective measurements about the patient's health status [37].

### 1.4. Predictors of Hiring Paid Care Workers

From a practical level, there have been few quantitative studies that investigated the predictors of using home care services or the hiring of live-in MCWs. Rogero-García and Rosenberg [38], in a study focused on Spain, found that having an informal caregiver with a higher education level, higher family income, and having a smaller household size increased the odds of receiving paid care support. Pego and Nunes [39] found in the Portuguese context that people in need of care that had higher levels of formal education, lived without a spouse, and had a chronic health condition were more likely to receive home care from a paid care worker.

In the Italian context, Di Rosa et al. [40] found that the severity of the person in need of care's disability, having an informal caregiver that lived close to but not with the person in need of care, and if the informal caregiver was currently working increased the odds of hiring an MCW. Another study found that individuals that used public formal care services had increased odds of hiring an MCW [41]. Barbabella et al. [19], in a study focused on Alzheimer's patients in the Marche region, found that higher levels of education among caregivers and receiving the IdA increased the odds of hiring an MCW. The latter finding was related to the fact that the IdA increased the amount of resources available to the care recipient's household or to that of their primary caregiver, thus making it more likely that they can afford to hire an MCW.

### 1.5. Objectives and Aims

Italy's ageing population, reliance on live-in MCWs to provide LTC, and a lack of alternative formal care services raise some concerns around equality in access of care for the estimated 2.6 million older adults with LTC needs in Italy, especially for those who are not able to afford to hire a live-in MCW or do not receive informal care [7,10,11,42]. Ensuring that all adults can access affordable quality care in the setting they prefer is a crucial step in the development of a sustainable care market in Italy [43]. Likewise, this will also contribute to meeting the Sustainable Development Goal (SGD) target 3.8 on achieving access to quality essential healthcare services [44].

Concurrently, it is also essential that the Italian MCW market takes a high road to care work approach, where MCWs have decent employment and living conditions, enjoy a safe work environment, have access to adequate skills training or recognition of prior learning systems, and that their work is valued and fairly remunerated [45]. This will assist in ensuring that MCWs have improved migration outcomes and experiences and that SDG target 8.8 on protecting labour rights and the promotion of safe and secure working environments for all workers, including migrants, and, especially, women migrants; target 5.2 on eliminating all forms of violence against women in public and private spheres; and target 4.3 on ensuring equal access to affordable quality technical, vocational, and tertiary education are met [46–48].

To meet these goals, several research gaps need addressing. First, concerning the socio-economic predictors of hiring live-in MCWs among older adults with long-term care needs. The BH model details that the individual financing of health care plays an important role in access to health care services [37]. Barbabella et al. [19] noted that receiving the IdA is an enabling factor in hiring live-in MCWs in the Marche region of Italy. However, there may be other enabling factors not covered by this study that influence the odds of hiring MCWs, including the individual and household income of the person in need of care and if they have access to social security benefits. Second, studies by Barbabella et al. [19], Meyer [8], and Rugolotto, Larotonda, and van der Geest [21] showed that there might be some financial barriers to hiring live-in MCWs. However, to our knowledge, there is currently no data available on if those that hire a live-in MCW have difficulties affording this cost. Lastly, while there has been extensive qualitative research that detailed the

around the clock nature of live-in care work in Italy (see: [22,29,35,49,50]), there was little quantitative data on the number of hours of care provided by MCWs, the care tasks they perform, or people in need of care's views on the care provided.

Consequently, and in the context of the Marche region of Italy, this study had three main aims:

1. To determine the predictors of hiring a live-in MCW among adults with LTC needs;
2. to analyse the costs and any financial barriers associated with the hiring of live-in MCWs by people in need of care and their informal caregivers;
3. to investigate the objective care burden, tasks provided, and perception of care provided by live-in MCWs.

## 2. Materials and Methods

### 2.1. Research Design

This article draws on baseline data from the study "the perspective of older people with LTC needs and their family caregivers in the Marche region" [51]. This was conducted in an area that, in terms of availability and use of LTC services like home care and residential care, lays in the middle of an ideal ranking of Italian regions, and therefore could be useful to understand the specificity of the Italian case. An overall report on the methodology and the main descriptive findings emerging from the study was published; however, the findings presented in this article have not previously been presented [51]. This was a longitudinal survey that aimed to provide an overview of the use of social-health services and interventions by older adults with LTC needs in this Italian region. The survey included both people in need of care and their primary informal caregivers.

The survey was conducted in all 13 districts of the Marche region (see Figure 2). To ensure the highest correspondence between the sample selected for this study and the universe of older people with LTC needs living in the Marche region, the share of the population over the age of 75 with severe limitations in performing usual activities was estimated for each of the 13 districts. This was stratified by age (using three groups: 75–79, 80–84, and 85+) and gender. The resulting data were then used as a basis to identify the number of respondents to be interviewed in each district. The family caregiver most involved in providing daily support (i.e., the primary caregiver) was asked to respond to the questions concerning informal care-related issues.

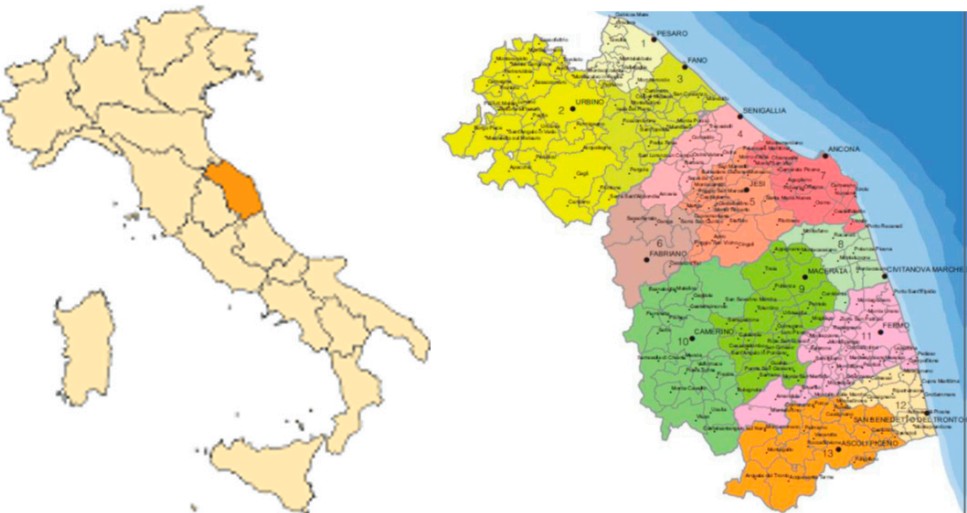

**Figure 2.** Location of the Marche region within Italy and its 13 health districts [52,53].

Due to difficulties in reaching some areas, a light overrepresentation emerged for two districts (San Benedetto and Ascoli), while three others (Fano, Fabriano, and Ancona) were

underrepresented. This, however, did not impact the overall representativeness of the study for the region taken as a whole as deviations in some areas were offset by others.

### 2.2. Eligibility Criteria for Inclusion

To determine eligibility for the survey, potential participants were asked two pre-screening questions. The first asked if they received IdA. Those receiving the allowance automatically met the inclusion criteria, while those that did not were asked a series of questions aimed at determining their ability to carry out Activities of Daily Living (ADL) and Instrumental Activities of Daily Living (IADL). ADL questions were based on the Barthel Index [54], with IADL questions based on the scale developed by Fillenbaum [55]. The Barthel index is an internationally validated and widely used tool to evaluate an individual's ability to carry out various basic activities of daily life (ADL) [54]. This includes the ability to feed yourself, take care of yourself, move around the house, dress, manage personal hygiene including washing yourself, and the absence of urinary and/or faecal incontinence. A higher overall score shows a higher degree of self-sufficiency [56]. The instrument by Fillenbaum aims to identify impaired functional activity among older adults based on the ability to complete five types of IADL tasks. This includes if they can get to places out of walking distance, go shopping, prepare their own meals, do their own housework, and manage their own money [55,57].

Potential participants that scored low on the ADL-IADL scale were eligible to take part and asked to identify their primary caregiver to also take part in the survey. In a few instances, where the person in need of care either did not have or did not identify a primary informal caregiver, a paid home care worker was included as the second part of the dyad.

### 2.3. Recruitment and Data Collection

Participants were identified and recruited by means of the pensioner trade unions (SPI–CGIL (Sindacato Pensionati Italiani–Generale Italiana del Lavoro), FNP–CISL (La Federazione Nazionale Pensionati–Confederazione Italiana Sindacati Lavoratori), and UILP (L'Unione Italiana Lavoratori Pensionati)) in the Marche region of Italy. At the baseline, surveys were conducted in person by staff members of the trade unions who received ad hoc training on how to administer the data collection tool. Separate questionnaires in the Supplementary Materials were used for people in need of care and informal caregivers. Data collection for the baseline was conducted between November 2019 and March 2020. During the administration of the survey, the staff conducting the survey determined if the person in need of care was in a position to fill in the survey themselves. In about half of the cases, the person in need of care was not able to fill in the survey unassisted and the primary caregiver filled in the survey as a proxy for them for all questions not requiring a subjective response.

### 2.4. Study Sample

The study sample consisted of 366 informal caregivers—people in need of care dyads. The majority of both people in need of care (72.1 percent) and informal caregivers (64.2 percent) were women (see Tables 1 and 2). The mean age for people in need of care was 86.08 (±6.32) and 63.23 (±11.30) for informal caregivers. People in need of care had lower levels of formal education than informal caregivers. Only 9.9 percent of people in need of care had a high school or post-high school degree, compared to informal caregivers at 56.5 percent. The majority of people in need of care were either married (35.5 percent) or widowed (59.3 percent). Around two in three informal caregivers were married (69.1 percent). Most informal caregivers were caring for their parent (63.4 percent).

**Table 1.** Characteristics of informal caregivers, with and without hiring a live-in MCW (n = 366).

|  |  | No MCW<br>N = 88 | MCW<br>N = 278 | Total [a]<br>N = 366 | *p*-Value [b] |
|---|---|---|---|---|---|
| **Predisposing characteristics** | | | | | |
| Gender: women | | 64.0 | 64.8 | 64.2 | 0.899 |
| Age | | 63.56 (11.92) | 62.18 (9.04) | 63.23 (11.30) | 0.319 |
| Relationship status | | | | | 0.509 |
| | Married | 67.3 | 75.0 | 69.1 | |
| | Separated | 5.4 | 5.7 | 5.5 | |
| | Widowed | 10.4 | 6.8 | 9.6 | |
| | Other | 16.9 | 12.5 | 15.8 | |
| Education | | | | | <0.001 |
| | No title | 1.8 | 0.0 | 1.4 | |
| | Primary | 17.3 | 6.8 | 14.8 | |
| | Middle school | 29.9 | 19.3 | 27.3 | |
| | High school | 42.1 | 48.9 | 43.7 | |
| | Above high school | 9.0 | 25.0 | 12.8 | |
| Relationship to the person in need of care | | | | | 0.001 |
| | Husband/Wife | 25.9 | 4.5 | 20.8 | |
| | Son/Daughter | 60.4 | 72.7 | 63.4 | |
| | Other | 13.7 | 22.7 | 15.9 | |
| **Enabling factors** | | | | | |
| Distance to person in need of care | | | | | 0.001 |
| | Live in the same apartment | 67.6 | 17.0 | 55.5 | |
| | Live in same apartment building | 11.2 | 17.0 | 12.6 | |
| | Walking distance | 11.5 | 33.0 | 16.7 | |
| | Need to travel by car, bus, or train | 9.8 | 32.9 | 15.0 | |
| Currently working [c]: yes | | 30.4 | 35.6 | 31.7 | 0.364 |

Notes: MCW—Migrant care worker. [a]—mean or percentage as appropriate. [b]—results of chi square test for categorical variables and two-tailed t-test for continuous variables. [c]—n = 276 for no MCW, 87 for MCW, and 363 for total.

**Table 2.** Characteristics of people in need of care, with and without hiring a live-in MCW (n = 366).

|  |  | No MCW<br>N = 88 | MCW<br>N = 278 | Total [a]<br>N = 366 | *p*-Value [b] |
|---|---|---|---|---|---|
| **Predisposing characteristics** | | | | | |
| Gender: women | | 69.8 | 79.5 | 72.1 | 0.075 |
| Age | | 85.37 (6.13) | 88.32 (6.44) | 86.08 (6.32) | <0.001 |
| Relationship status | | | | | 0.039 |
| | Married | 39.6 | 22.7 | 35.5 | |
| | Widowed | 55.8 | 70.5 | 59.3 | |
| | Other | 4.7 | 6.8 | 5.2 | |
| Education level [c] | | | | | 0.143 |
| | No title | 13.1 | 13.8 | 13.3 | |
| | Primary | 64.7 | 64.4 | 64.6 | |
| | Middle school | 13.5 | 8.0 | 12.2 | |
| | High school | 5.5 | 11.5 | 6.9 | |
| | Above high school | 3.3 | 2.2 | 3.0 | |
| **Enabling factors** | | | | | |
| *Individual level* | | | | | |
| Work pension [d]: yes | | 71.1 | 63.6 | 69.3 | 0.185 |
| Social pension: yes | | 10.8 | 19.3 | 12.8 | 0.037 |
| Disability pension: yes | | 16.5 | 15.9 | 16.4 | 0.888 |

**Table 2.** *Cont.*

|  |  | No MCW N = 88 | MCW N = 278 | Total [a] N = 366 | *p*-Value [b] |
|---|---|---|---|---|---|
| Cash-for-care allowance (IdA): yes |  | 78.8 | 90.9 | 81.7 | 0.010 |
| Survivor's pension [d]: yes |  | 46.6 | 61.4 | 50.1 | 0.016 |
| Annuities or income from rent/interest/dividends: yes |  | 5.0 | 11.4 | 6.6 | 0.037 |
| Income [c] |  |  |  |  | 0.030 |
|  | Less than 500 EUR/month | 1.8 | 1.1 | 1.6 |  |
|  | 500–1000 EUR/month | 27.7 | 14.9 | 24.6 |  |
|  | 1000–1500 EUR/month | 51.3 | 56.3 | 52.5 |  |
|  | 1500–2000 EUR/month | 15.3 | 14.9 | 15.2 |  |
|  | More than 2000 EUR/month | 4.0 | 12.6 | 6.1 |  |
| *Contextual level* |  |  |  |  |  |
| Household income [e] |  |  |  |  | 0.561 |
|  | Less than 1000 EUR/month | 5.3 | 2.4 | 4.7 |  |
|  | 1000–1500 EUR/month | 24.0 | 32.9 | 26.2 |  |
|  | 1500–2000 EUR/month | 24.0 | 24.4 | 24.1 |  |
|  | 2000–2500 EUR/month | 22.5 | 17.1 | 21.2 |  |
|  | 2500–3000 EUR/month | 14.9 | 14.6 | 14.8 |  |
|  | More than 3000 EUR/month | 9.2 | 8.5 | 9.0 |  |
| **Needs characteristics** |  |  |  |  |  |
|  | ADL | 43.00 (25.53) | 34.94 (21.34) | 41.07 (23.25) | 0.004 |

Notes: MCW—Migrant care worker; ADL—Activities of Daily Living; IdA—Indennità di accompagnamento; [a]—mean or percentage as appropriate; [b]—results of chi square test for categorical variables and two-tailed t-test for continuous variables; [c]—n = 275 for no MCW, 87 for MCW, and 362 for total; [d]—n = 277 for no MCW, 88 for MCW, and 365 for total; [e]—n = 262 for no MCW, 82 for MCW, and 344 for total.

Around one in four (24.0 percent) people in need of care hired a live-in MCW. Concerning the characteristics of the hired MCWs, almost all were women (97.7 percent), and half held the nationality of a country within the European Union. The mean age of the MCWs was 51.76 ($\pm$9.56).

*2.5. Measures*

Participants were asked, among other things, which home care services they were provided from paid care workers. This included both live-in or live-out care workers. People in need of care were then asked how many months they had used this service, the total number of hours the worker was employed per week, the monthly costs spent on paid care services, if the worker had received skills training specifically around caring for older adults or people with disabilities, and also the nationality, gender, and age of the care worker. The person in need of care and the primary informal caregiver were also asked if they had had or would have had difficulties affording the costs of hiring a care worker over the past year. Moreover, people in need of care were also asked their thoughts on the care they received from an MCW.

Predisposing characteristic measures included the gender of the person in need of care as well as their relationship status, age, and level of education. Predisposing characteristics

for the primary caregiver included their gender, age, relationship status, level of education, and relationship status to the person in need of care.

Enabling factors at the individual level included the person in need of care's access to social services, including work pensions, social pensions (only available to those over 67 and with a net income of less than EUR 5954 per year), disability pension, IdA, survivor's or indirect pension, other forms of income or assets, care allowances administered from the municipality, or care allowances from the region. Moreover, the income levels of the person in need of care were also reported. Likewise, this study also measured if the primary caregiver was currently working and the distance between the person in need of care and their family caregiver. This included if the two lived together, lived in the same building, lived within walking distance from each other, or if the primary caregiver needed to travel by train, car, or bus to reach the person in need of care. Contextual enabling factors included household income levels. Needs characteristics included the Barthel scale.

*2.6. Data Analysis*

As the primary aim of this study was to determine the predictors of hiring an MCW among older adults with LTC needs, variables tested in this article were based on a combination of factors identified in the BH model and from past studies on these predictors. As previous research showed that factors relating to both the person in need of care and their primary caregiver influenced the odds of hiring an MCW; factors from both actors were included in this study.

Univariate analyses were used to provide an overview of the costs associated with hiring an MCW, the number of hours of care provided by MCWs, and if the person in need of care and/or their primary caregiver had any difficulties in affording to hire an MCW.

The nationality of the MCW was re-categorised into those that have the nationality of a country within or outside the European Union. People in need of care's thoughts on the care provided by an MCW were re-categorised into if the response was positive, positive and negative, neutral, or negative.

Descriptive analysis, including univariate and bivariate analysis, was used to compare any significant differences between people in need of care that hired an MCW and those that did not. For categorical variables, a chi-square test was used to test significance (*p*-value less than 0.05), while a two-tailed t-test was used for continuous variables.

To avoid issues of collinearity between variables, a chi-square test was used to test for collinearity between categorical variables (*p*-value less than 0.05). To test for collinearity with continuous variables, the Variance Inflation Factor (VIF) and tolerance were used based on running a linear regression with the dichotomous variable as the dependent variable and continuous variables as independent variables.

For all analysis, dyads that included a person in need of care and a paid care worker were excluded. Likewise, cases were also excluded when the person in need of care hired an Italian live-in care worker, as similar predictors may be found between hiring a live-in care worker (migrant or Italian).

Binary logistic regression was used to calculate the predictors of hiring an MCW (dependent variable). Binary logistic regression rather than multi-level logistic regression was used, due to the survey not having representative data available on all districts in the Marche region. For logistic regression, some variables, including the income of the person in need of care and if the person in need of care lived with their primary caregiver were excluded, as there was collinearity with other independent variables.

For the binary logistic regression, the primary caregiver education variable was dichotomised to either having a high school education and higher or below. Variables were only included in the regression if they were statistically significant at the bivariate analysis level (*p*-value less than 0.05).

## 2.7. Ethics

The study was submitted for ethics committee approval at the National Institute of Health and Science on Ageing (INRCA), Italy. However, since the investigation did not imply the involvement of clinical patients, this committee deemed clearance in this regard as not necessary.

## 3. Results

### 3.1. Characteristics of Those That Hired or Did Not Hire a Live-In MCW

3.1.1. Socio-Demographic Factors and Health Status

Table 1 provides an overview of predisposing characteristics and enabling factors of informal caregivers, based on if they provide care for someone that hired or did not hire a live-in MCW. Likewise, Table 2 provides a summary of predisposing characteristics, enabling factors, and needs characteristics of people in need of care, based on if they did or did not hire a live-in MCW.

Regarding the predisposing characteristics of the informal caregivers, there were no major differences in gender for those that cared for someone that hired (64.8 percent women) or did not hire (64.0 percent women) an MCW. A higher proportion of those that cared for someone that hired an MCW were married (75.0 percent) compared to those that did not (67.3 percent). However, this result was not statistically significant.

Around three-quarters of informal caregivers that cared for a person in need of care that hired an MCW had a high school degree or above (73.9 percent), compared to only 51.1 percent that did not. A higher proportion of those that cared for someone that did not hire an MCW were the husband or wife of the person in need of care (24.5 percent), compared to only 4.5 percent of those that did.

Regarding enabling factors of informal caregivers, only 17.0 percent that cared for someone that hired an MCW lived with the person in need of care, compared to two in three that did not hire an MCW (67.6 percent).

Concerning predisposing characteristics of people in need of care, 79.5 percent that hired an MCW worker were women compared to 69.8 percent that did not hire an MCW. However, this result was not statistically significant. A higher proportion of those that hired an MCW were widowed (70.5 percent) compared to those that did not (55.8 percent). There were no large differences in age or education level.

For enabling characteristics of the person in need of care, 12.6 percent of those that hired an MCW had an income of over EUR 2000 a month compared to 4.6 percent of those that did not hire an MCW. There were no major differences regarding the levels of household income.

With respect to the characteristics of the needs of the person in need of care, those that hired an MCW ($34.94 \pm 21.34$) had lower ADL scores than those that did not ($43.00 \pm 25.53$).

3.1.2. Support Services

In terms of care services and allowances, a higher percentage of those that hired an MCW received a social pension (19.3 percent), IdA (90.9 percent), survivor's pensions (61.4 percent), and annuities (11.4 percent), compared to those that did not hire an MCW (10.8, 78.8, 46.6, and 5.0 percent, respectively).

In contrast, those that hired an MCW (63.6 percent) were less likely to receive a work pension than those that did not hire an MCW (71.1 percent). No major differences were reported concerning disability pensions.

### 3.2. Predictors of Hiring Live-In MCWs

A bivariate analysis of the predictors associated with the hiring of a live-in MCW is shown in Table 3. For predisposing characteristics, the age of the person in need of care was statistically significant but did not lead to a large increase in the odds of hiring an MCW (Odds Ratio (OR) = 1.077; 95% Confidence Interval (CI): 1.034–1.122; $p$-value < 0.001). Having a primary caregiver that had a high school education or above significantly increased

the odds that the person in need of care would hire a live-in MCW (OR = 3.880; 95% CI: 1.982–7.595; *p*-value < 0.001).

**Table 3.** Predictors associated with the hiring of a live-in MCW (n = 368 dyads).

| | OR | 95% CI | | *p*-Value |
|---|---|---|---|---|
| **Person in need of care** | | | | |
| *Predisposing characteristics* | | | | |
| Age (1 year increase) | 1.077 | 1.034 | 1.122 | <0.001 |
| *Enabling characteristics* | | | | |
| Receiving social pension (ref: no) | 2.258 | 1.121 | 4.549 | 0.037 |
| *Needs characteristics* | | | | |
| ADL (1 point increase) | 0.982 | 0.971 | 0.994 | 0.005 |
| **Informal caregiver** | | | | |
| *Predisposing characteristics* | | | | |
| Level of formal education (ref: below high school diploma) | 3.880 | 1.982 | 7.595 | <0.001 |

Notes: OR—Odds Ratio; CI—Confidence interval; ADL—Activities of Daily Living.

Regarding enabling characteristics, people in need of care that received a social pension had increased odds (OR = 2.258; 95% CI: 1.121–4.549; *p*-value: 0.037) of hiring an MCW, compared to those that did not receive this pension.

For needs characteristics, the ADL score of the person in need of care was statistically significant but did not have a large influence on the odds of hiring an MCW (OR = 0.982; 95% CI: 0.971–0.994; *p*-value: 0.005).

*3.3. Characteristics of Care Provided*

Table 4 highlights that, on average, people in need of care had been receiving care from an MCW for almost three years, i.e., 35.43 (±40.65) months. Only 6.8 percent of people in need of care reported that the MCW they hired had received specific skills training for caring for older adults or someone with a disability.

**Table 4.** Characteristics of care provided by live-in MCWs.

| | Total [a] |
|---|---|
| Number of months being provided care by an MCW (n = 84) | 35.43 (40.65) |
| The MCW had received skills training specific to caring for older adults or someone with a disability (n = 88) | |
|     Yes | 6.8 |
|     No | 68.2 |
|     Do not know | 25.0 |
| Person in need of care's views on the care provided by MCWs (n = 88) | |
|     Positive | 89.8 |
|     Positive and negative | 3.4 |
|     Neutral | 5.7 |
|     Negative | 1.1 |
| Hours of care provided per week (n = 85) | 75.22 (48.16) |
| Worked above 54 h per week [b]: yes (n = 85) | 37.5 |
| Care tasks provided/hours per week | |
|     Personal care and hygiene (n = 80) | 33.59 (42.47) |
|     Home mobility (n = 71) | 24.45 (30.91) |
|     Supervision (n = 72) | 53.08 (54.29) |
|     Domestic work including cooking (n = 68) | 28.12 (30.00) |
|     Mobility outside the home (including transportation; n = 39) | 22.86 (36.48) |

Notes: MCW—Migrant Care Worker; [a]—percentage or mean as appropriate; [b]—the number of hours per week as set out in the cooperative bargaining agreement for care and domestic workers.

The average hours of care provided by an MCW were 75.22 (±48.16) per week. This number is likely an overestimate due to some participants indicating that the MCW worked 168 h a week. Around one in three (37.5 percent) MCWs worked more than 54 h a week.

MCWs spent most of their time on supervision tasks (53.08 ± 54.29 h/week), followed by personal care and hygiene (33.59 ± 42.47 h per week). Similar to the total number of hours of care, individual care tasks were also skewed due to the perception of providing round-the-clock care, as is shown by the high standard deviation.

People in need of care were also asked their thoughts about the care they received. Nine in ten (89.8 percent) participants responded positively to the care provided by the MCW. Common responses included that the care provided was good, indispensable, or helpful: 3.4 percent of participants responded both positively and negatively; 5.7 percent gave neutral responses, with two participants citing the need for increased training for the MCW; only 1 participant (1.1 percent) responded negatively.

### 3.4. Financial Aspects of Care Provision

Table 5 shows that approximately half of the people in need of care had (57.5 percent) or would have had (49.3 percent) difficulties affording the costs of hiring an MCW in the past year. Around one in four primary caregivers (27.6 percent) that provided care for a person in need of care that hired an MCW had a lot of problems resulting from the additional expenses of hiring an MCW, with a further 28.7 percent often experiencing problems. Around three in four (71.3 percent) informal caregivers that provided care for a person in need of care that hired an MCW contributed to the costs of hiring the MCW.

**Table 5.** Costs and financial difficulties resulting from hiring a live-in MCW.

|  | No MCW | MCW | Total [a] |
|---|---|---|---|
| Did the person in need of care have or would have had difficulties in the past year in affording the costs of hiring a live-in MCW? | n = 272 | n = 87 | n = 359 |
| yes | 49.3 | 57.5 | 51.3 |
| Did the informal caregiver have or would have had problems in the last year relating to the additional expenses resulting from the hiring of a live-in MCW? | n = 276 | n = 87 | n = 363 |
| No additional expenses | 66.3 | 28.7 | 57.30 |
| Never | 11.6 | 4.6 | 9.9 |
| Sometimes | 6.5 | 10.3 | 7.4 |
| Often | 9.8 | 28.7 | 14.3 |
| A lot | 5.8 | 27.6 | 11.0 |
| The cost spent per month on hiring an MCW in EUR (n = 81) | N/A | 1167.46 (247.90) | 1167.46 (247.90) |
| The MCW received on or above the minimum wage for those caring for an older adult with long-term care needs [b]: yes (n = 81) | N/A | 76.1 | 76.1 |

Notes: MCW—Migrant Care Worker; N/A—Not Applicable; [a]—percentage or mean as appropriate; [b]—983.22 EUR/month in 2019 and EUR 984.02 in 2020.

The average cost per month spent on hiring an MCW was EUR 1167.46 (±247.90). Approximately one in four (23.9 percent) MCWs were earning less than the minimum wage stipulated for care workers providing care for someone with LTC needs.

## 4. Discussion
### 4.1. Links to the Existing Literature

This article aimed to determine the socio-economic predictors of hiring a live-in MCW among older adults with LTC needs in the Marche region of Italy. The findings of this study were largely consistent with previous studies on the predictors of hiring MCWs or paid care workers. Regarding predisposing characteristics, people in need of care that

had an informal caregiver with a higher level of formal education had increased odds of hiring a live-in MCW. This result was similar to findings from previous studies by Barbabella et al. [19] and Rogero Garcia and Rosenberg [38]. Several factors might explain this phenomenon. First, it might be that informal caregivers with higher levels of formal education increased awareness of the care needs of the person in need of care [19]. Second, while this study did not find that an increased household income increased the odds of hiring an MCW, this did not include the income of informal caregivers that did not live with the person in need of care. This study found that around three in four informal caregivers contributed to the costs of hiring the MCW. Consequently, assuming that higher levels of formal education led to the increased likelihood of higher income levels, hiring an MCW would therefore be more affordable [19]. Third, while the informal caregiver's level of formal education was statistically significant, this was not the case for the level of education of the person in need of care. This could be reflective of the organisational role that informal caregivers take in the management of MCWs. Previous research by Scrinzi [49] and Gallo and Scrinzi [58] found that it was often informal caregivers, and not the persons in need of care themselves, that were responsible for the hiring of the MCW.

For enabling characteristics, receiving a social pension increased the odds of hiring an MCW. This pension was only available to adults over the age of 67 that have an annual income of less than EUR 5954 per year. Consequently, this might indicate that individuals with lower levels of income were reliant on this payment to be able to afford to hire an MCW.

For needs characteristics, unlike Di Rosa et al. [40] and Pego and Nunes [39], this study did not find that a higher level of care needs was associated with increased odds of hiring an MCW. This might be because having LTC needs was an eligibility requirement for participation in the study. Therefore, incremental changes in ADL scores might not have led to increasing the odds of hiring an MCW. This might also explain why increases in the age of the person in need of care also did not significantly increase the odds of hiring an MCW.

This article also aimed to analyse the costs and any financial barriers associated with the hiring of live-in MCWs. The findings in this article highlighted that many people in need of care and their primary informal caregivers experienced difficulties affording or had problems resulting from the costs of hiring an MCW. This could indicate that the costs necessary to hire an MCW were either unaffordable or were a financial burden for many [10,19,21]. Consequently, because hiring an MCW through formal channels is not affordable, some people in need of care and their family members might hire workers outside the formal sector, which reduces the labour rights afforded to migrant workers [21]. While this study did not ask about the migration status of the hired MCWs, the fact that some of the workers earned below the salary indicated in the collective bargaining agreement for MCWs, might suggest that some workers did not have registered employment contracts or visas/residency permits.

Socio-economic barriers to hiring an MCW are concerning in the case of Italy due to the over-reliance on either MCWs or informal caregivers to provide home care. Moreover, a study by Tur-Sinai et al. [11] found that people in need of care that come from lower-income families were less likely to receive informal care. Consequently, situations might arise where a person in need of care does not have an informal caregiver to provide care for them nor is able to afford to hire a paid care worker.

This study also aimed to investigate the objective care burden, tasks provided, and perception of care provided by live-in MCWs. The results of this study showed that MCWs were shouldering a high degree of care burden, as evidenced by the high number of reported hours of care they provided per week. This was likely representative of the views of many informal caregivers and people in need of care that MCWs should be expected to work around the clock, even if this did violate the conditions set out in the collective bargaining agreement for MCWs [22,28,29,49,50]. The excessive number of hours that MCWs worked might also be a reflection of the undervaluing of care work present in

Italian society as well as asymmetrical power relationships between MCWs and people in need of care and their family members [6,8,22,29,49,59]. Likewise, these power imbalances might also explain why MCWs worked long hours despite people in need of care reporting largely positive experiences with the services provided by MCWs. Alternatively, or in addition to the above points, the long hours might also be reflective of the lack of alternative LTC services available to people in need of care [7].

The undervaluing of care work might also explain why so few MCWs had received any sort of skills training related to caring for older adults or for the specific care needs of the person in need of care. Previous research by Scrinzi [49] showed that many employers of MCWs had the assumption that women are natural caregivers and, therefore, do not place a large value on the skills necessary to provide care work. This is problematic on several fronts. First, while not covered explicitly by this study, this undervaluing of care work and lack of skills training or recognition of prior learning (RPL) could result in MCWs having few opportunities to advance in their career [60]. Second, this might also lead to people in need of care receiving a low level of quality of care, as MCWs may not receive training specific to their health condition(s) or care needs.

### 4.2. Implications for Policy and Practice

There are several steps that the Italian government can take to help reduce the socio-economic barriers present in the Italian MCW market, improve working conditions for MCWs, increase the quality of care for people in need of care, and consequently improve the sustainability of the Italian MCW market. First, increasing the regulation of the IdA by making it a requirement that people hire paid care workers with regular contracts might assist in increasing the formality of the Italian MCW market [10,18,19]. The Home Care Premium scheme already showed that requiring people in need of care to hire workers through formal channels and with registered contracts was an effective way to increase formality in the sector [18]. This also proved to be effective in the Netherlands and France, which also applied restrictions to cash-for-care schemes [3,45,61,62]. Increasing the number of MCWs with formal employment status would help to reduce some of the precariousness present in the sector and to minimise the gap between the formal and informal care market [10]. Likewise, it might also assist in meeting SDG target 8.8 on protecting labour rights and the promotion of safe and secure working environments for all workers, including migrants, and, especially, women migrants [48].

However, increased regulation of the IdA is unlikely to be a sustainable long-term solution if hiring an MCW is not affordable for people in need of care and their family members. It is clear from this study that many people in need of care struggled to afford to hire an MCW. Therefore, increased funding is needed for people in need of care [10]. One possible solution would be to add a means-based assessment to the IdA or other allowances, as this might assist in identifying people most in need of financial assistance [19]. Alternatively, or in addition to this, tax breaks for those that hire paid care workers might also assist in reducing the costs for people in need of care [10,63]. Increasing the affordability of the care market would also aid in meeting SDG target 3.8 on achieving access to quality essential healthcare services [44].

It is also important that Italy moves away from viewing care work as low or unskilled work. For this to be possible, it would require the exposure of existing racialised and gendered ideologies around care and migration [49]. Education campaigns run by the Italian government around the value of paid and unpaid care work might assist in increasing how society values care work and might, therefore, reduce the number of hours MCWs are expected to work [45]. Likewise, it might also offer an avenue to meeting SDG target 5.4 on recognizing and valuing unpaid care and domestic work; therefore, assisting not only MCWs but also informal caregivers [47].

To increase the quality of care provided and to provide MCWs with more long-term career opportunities, the Italian government must prioritise skills recognition and recognition of prior learning (RPL) systems [60]. This would help with reaching SDG

target 4.3 on ensuring equal access to affordable quality technical, vocational, and tertiary education. In Italy, few MCWs receive any forms of skills training, and those who do have some sort of relevant professional qualifications often failed to have these recognised [49]. In 2019, the Italian government introduced a qualification system for care workers (see UNI 11766/2019 standard); however, there has been a low level of implementation of this system so far [10]. Some regions also provide skills training for MCWs; however, while this is a positive step, a national-level system is also necessary [42].

The home care setting can be an isolating experience for MCWs and is often characterized by a lack of enforcement of labour rights [64]. Additional labour rights inspections provide one avenue to determine if MCWs are working excessive hours or are experiencing labour rights abuses or violations. These inspections should take place separate from immigration procedures and should be in line with the International Labour Organization (ILO) Labour Inspection Convention (No. 81) [65]. Similar to increasing the formality of the MCW market through increased regulation, labour rights inspections offer an avenue for achieving SDG target 8.8 on protecting labour rights and the promotion of safe and secure working environments of all workers, including migrants, and, especially, women migrants, and target 5.2 on eliminating all forms of violence against women in public and private spheres [47,48].

### 4.3. Limitations

There were some limitations to this article that should be considered to interpret the reported findings. First, this article only used baseline data. Consequently, the results were cross-sectional and only showed interactions between variables and did not show causality. Likewise, as this article mainly focused on the individual determinants of access to care, there were likely to be some external factors that also influenced the odds of hiring a live-in MCW. The BH model highlights that contextual factors should also be included when analysing health service usage, while the only contextual factor used in the logistic regression in this article was household income [37]. Other external variables that were not included in this article but might be useful to analyse in future studies include family income levels and cultural preferences around willingness to use health services both at the individual and contextual level. Likewise, because health care systems in Italy are run at the regional level, further research is needed in other Italian regions to see how these differences in services influence the odds of hiring an MCW [18].

Also, this study did not measure the specific health conditions of the person in need of care, but only their functional impact. While this study did include the overall care needs of the person receiving care, it might be that having specific health conditions that require an increased level of care or specific care skills might influence the odds of hiring an MCW. There might also be some bias in the study sample due to how data was collected. Potential participants were identified through pensioner trade unions. Consequently, these individuals might have higher levels of support or social relationships than those that did not have contact with unions. Moreover, during data collection, some districts in the Marche region were under-represented in the study sample. As these districts have different geographic areas, availability of services, and urban and rural cultural differences, this might influence the odds of service use [19].

### 5. Conclusions

Despite these limitations, this study built upon and addressed research gaps from past studies to update and enhance our understanding of the socio-economic factors that influenced the recruitment of live-in MCWs in Italy, the objective care burden placed on MCWs, and the financial barriers that people in need of care and informal caregivers face when hiring MCWs.

The combination of findings presented in this article pointed to several areas of concern regarding the sustainability of the Italian MCW. These included potential barriers for informal caregivers with low levels of formal education to facilitate the hiring of a

live-in MCW for the person they care for, financial difficulties resulting from hiring an MCW for people in need of care and informal caregivers, low levels of formal skills training related to care work for MCWs, and high objective care burdens and wages below what was outlined in the collective bargaining agreements for MCWs.

Addressing each of these issues would help to ensure equality in access to and high-quality care for older adults with LTC needs and decent work opportunities for MCWs. While this study only focused on the Marche region of Italy, the results might provide insight into the situation of the estimated 2.6 million older adults with LTC needs in Italy and partly also offer reflections for many other countries in which the phenomenon of hiring live-in MCWs is widespread [10].

**Supplementary Materials:** The following are available online at https://www.mdpi.com/article/10.3390/su13105349/s1. Questionnaire S1 for person in need of care; Questionnaire S2 for informal caregiver.

**Author Contributions:** Conceptualisation: G.L.; methodology: G.L. and P.F.; formal analysis: O.F. and P.F.; investigation: G.L.; data curation: P.F.; writing—original draft preparation: O.F.; writing—review and editing: O.F. and G.L.; project administration: G.L.; funding acquisition: G.L. All authors have read and agreed to the published version of the manuscript.

**Funding:** This research has partially been funded by the European Union's Horizon 2020 research and innovation programme under the Marie Skłodowska-Curie grant agreement No 814072. This study has partially been supported by the Ricerca Corrente funding from the Italian Ministry of Health to IRCCS INRCA.

**Institutional Review Board Statement:** The study was submitted for ethics committee approval at the National Institute of Health and Science on Ageing (INRCA), Italy. However, since the investigation did not imply the involvement of clinical patients, this committee deemed clearance in this regard as not necessary.

**Informed Consent Statement:** Informed consent was obtained from all subjects involved in the study.

**Data Availability Statement:** Data is not yet publicly available, but is planned to be made available at the end of 2021.

**Acknowledgments:** Authors would like to acknowledge the substantial technical support provided by Flavia Piccinini to coordinate the collection of the data used for this study.

**Conflicts of Interest:** The authors declare no conflict of interest.

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
