# Peer review of "Socio-Economic Predictors of Hiring Live-In Migrant Care Workers to Support Community Dwelling Older Adults with Long-Term Care Needs: Recent Evidence from a Central Italian Region"

_sustainability, doi:10.3390/su13105349_

Round 1

Reviewer 1 Report

The baseline data from "the  perspective of older people with LTC needs and their family caregivers in the Marche regions" conducted by the pension organizations used in this paper is invaluable. This is because, although this was at the local level, there is no national level survey with such details.
Also, I think that the conceptual framework , research method and data analyses in this paper  are very academic and appropriate.

However, they are so diverse that they appear to be a bit inconsistent. Also,  in the 3.Results and 4. Discussion, It cannot be said that there was any findings that surpassed the previous studies regarding the Italian case. However, as mentioned above, the analysis using this survey provided a much clearer picture of the care recipients and their families hiring MCWs in Italy. Additionally, this study mentions the previous research in great detail and clarifies the role and position of this research.

Author Response

Dear reviewer,

Thank you very much for the feedback. We have prepared a table (attached) which shows the changes we have made based on your feedback. In case it is also useful, we have included the feedback from the other reviewer, as we believe some of the suggested changes also correspond to the feedback you provided. Please also note that the lines referred to in the action and response column refer to the track changes version of the article.

Reviewer 2 Report

Dear Authors,

I thoroughly read your paper and found it quite interesting. You are addressing a very important topic in the field of elder care against the demographic situation and development in Italy. Nonetheless, I fail to see the link(s) to the scope of the journal “Sustainability”. In the following, please find my comments.

  1. Introduction

Put line 48 to 53 at the beginning.

Sharpen and shorten sub-section 1.2

Sub-section 1.3 presents the Behavioural Model of Health Service Use. This sub-section is very important and should be a separate section.

Please revise sub-section 1.4. Point out the research gaps (see line 163ff), insert relevant sources and make clear the purpose of the paper.

Line 170 to 173 present results. This information need to be shifted to the “Results” section.

  1. Materials and Methods

Insert a paragraph on the research design and the methodological steps.

  • Data: As I understand for this paper data from a primary study is used (see line 177ff). Please insert the source and explain whether and what findings have been already published (elsewhere).

Please describe the study setting more precisely, (maybe) provide a map and explain the particularities of Marche region relating to elder care and caregivers’ support needs.

  • Measures: Please provide information on sampling and data collection.

Insert the questionnaire(s) (annex). Did you apply two different questionnaires for the cared-for people and the caregivers?

For “Sustainability” is not a medical or healthcare journal like “IJERPH” or “Healthcare” (both also MDPI) an explanation of The Barthel scale is necessary.

  • Data analysis: Please explain “VIF” (line 250).

  1. Results

This section is confusing and difficult to read.

Please present the empirical findings in a theme-centred manner.

Please rename the sub-headings and use theme-focussed titles.

This section presents facts. Consequently, assumptions and conclusions need to be avoided (line 279ff).

Table 1 is unreadable. Please, split into two tables – one for presenting the results for the “Person in need of care”, one for “Informal caregiver”). Insert a reference to Table 1 in the text. The same applies for Table 4.

  1. Discussion

This section lacks context to sustainability. Please, interlink your findings with the SDGs and the spatial context.

Insert a sub-heading above the paragraph which deals with the limitations of your study (line 448).

  1. This should be a separate section. Moreover, I would like to ask for more specification.

All the best!       

Author Response

Dear reviewer,

Thank you very much for the feedback. We have prepared a table (attached( which shows the changes we have made based on your feedback. In case it is also useful, we have included the feedback from the other reviewer, as we believe some of the suggested changes also correspond to the feedback you provided. Please also note that the lines referred to in the action and response column refer to the track changes version of the article.

Round 2

Reviewer 2 Report

Dear Authors,

I enjoyed reading the revised version of your paper and I am looking forward to the publication.

All the best!